# Performance and Mechanism of Chlorine Dioxide on BTEX Removal in Liquid and Indoor Air

**DOI:** 10.3390/molecules28114342

**Published:** 2023-05-25

**Authors:** Anlong Wang, Yina Qiao, Yufan Zhang, Riya Jin, Jiaoqin Liu, Zengdi He, Mengye Jia, Jingshuai Gao, Chengjie Guo

**Affiliations:** School of Environment and Safety Engineering, North University of China, Taiyuan 030051, China

**Keywords:** benzene, toluene, xylene, ClO_2_, semi-enclosed space, ab initio

## Abstract

With the development of the chemical industry, benzene, toluene, ethylbenzene, and xylene (BTEX) have gradually become the major indoor air pollutants. Various gas treatment techniques are widely used to prevent the physical and mental health hazards of BTEX in semi-enclosed spaces. Chlorine dioxide (ClO_2_) is an alternative to chlorine as a secondary disinfectant with a strong oxidation ability, a wide range of action, and no carcinogenic effects. In addition, ClO_2_ has a unique permeability which allows it to eliminate volatile contaminants from the source. However, little attention has been paid to the removal of BTEX by ClO_2_, due to the difficulty of removing BTEX in semi-enclosed areas and the lack of testing methods for the reaction intermediates. Therefore, this study explored the performance of ClO_2_ advanced oxidation technology on both liquid and gaseous benzene, toluene, o-xylene, and m-xylene. The results showed that ClO_2_ was efficient in the removal of BTEX. The byproducts were detected by gas chromatography-mass spectrometry (GC-MS) and the reaction mechanism was speculated using the ab initio molecular orbital calculations method. The results demonstrated that ClO_2_ could remove the BTEX from the water and the air without causing secondary pollution.

## 1. Introduction

As an important chemical raw material, benzene, toluene, ethylbenzene, and xylene (BTEX) are widely used in many industries, and their hazards are of increasing concern. Benzene was listed as a “known human carcinogen” (Group A) in the EPA’s 1986 risk assessment guidelines and was classified as a known human carcinogen in 2005 [1]. Overexposure to benzene causes not only symptoms such as headaches, dizziness, drowsiness, confusion, tremors and loss of consciousness, moderate eye irritation, and skin irritation, but also increases the risk of leukemia [2,3]. Toluene not only causes photochemical haze and ozone depletion due to its high photochemical reactivity, but is also toxic, mutagenic, and carcinogenic within the human body [4,5]. The most commonly reported symptoms, after chronic inhalation exposure to xylenes, are depression of the visual and central nervous system, and respiratory irritation. Irritability, depression, insomnia, agitation, extreme tiredness, tremors, impaired concentration, and short-term memory issues may occur after long-term occupational exposure to xylene [6,7,8]. As typical volatile organic compounds (VOCs), BTEX causes severe air pollution in semi-enclosed areas such as supermarkets, offices, residences, workshops, etc. [9,10].

In recent years, extensive research has been carried out on treating gaseous BTEX. The adsorption method consists of removing BTEX pollution by adsorbing the contaminants onto the surface of the adsorbent. It has sufficient history in design and operation, strong applicability, and can transform pollutants into valuable products [11,12]. Shim et al., modified the surface-wrinkled silica nanoparticles (WSNs) with organosilane, to adsorb gaseous BTEX. At room temperature, 50 mg of hexadecyltrimethoxysilane (HDTMS)-WSN can thoroughly remove 200 μg/m^3^ of benzene, toluene, and xylene mixed air samples [13]. Sui et al., investigated the removal and recovery of o-xylene using silica gel and a vacuum swing adsorption (VSA) process. Experiments showed that, at 5:1 for height to diameter ratio (H/D), 5 kPa (absolute) of desorption pressure, and 0.45 L/min of purge gas flow rate, the VSA was able to entirely remove o-xylene from the gas, with 87% of o-xylene recovered directly as a liquid [14]. Catalytic oxidation methods use catalysts to reduce the activation energy of the reaction so that BTEX can burn without flame until it is thoroughly degraded. Catalytic oxidation has a simple process, high treatment efficiency, and no secondary contamination compared to traditional thermal destruction methods [15,16,17]. Xiao et al., constructed a Pt/Ni-CeO_2_ catalyst via Ni doping. The result demonstrates that the conversion temperature of 90% toluene was reduced from 165 °C to 149 °C by the Pt/Ni-CeO_2_ catalyst compared with the Pt/CeO_2_ catalyst [18]. He et al., calcined CeO_2_ nanocubes at different temperatures. The study discovered that CeO_2_ nanocubes could fully catalyze the oxidation of o-xylene below 210 °C [19]. Chen et al., studied the removal of o-xylene using a novel Pd@NC/BN catalyst with a nitrogen-doped carbon layer, modified with Pd supported on hexagonal boron nitride. The results showed that the Pd@NC/BN catalyst has a removal rate of 90% for o-xylene at 185 °C [20]. Biological methods use o-xylene as energy and nutrients for microbial life activities, thereby degrading o-xylene through metabolism [10]. Biotreatment has the advantage of simple equipment, low operating costs, and less secondary pollution [16,21]. Dewidar et al., optimized the working conditions of the biotrickling filter for the mixed waste gases, toluene and styrene. The removal rate of both VOCs reached 90% with a total loading rate of 199.1 g/m and 90 s for empty bed residence times [22]. Li et al., constructed an airlift microbial electrolysis cell (AL-MEC) system to remove o-xylene. Under optimized conditions, while the inlet load of o-xylene was as high as 684 g/m^3^·h, the o-xylene removal rate remained at about 75% [23]. The technologies above can effectively remove BTEX, which has high industrial application value. Nevertheless, the requirement of these methods for particular operating conditions (e.g., high concentrations of BTEX) often makes such options less flexible, especially in semi-enclosed spaces. In this regard, the advanced oxidation removal of BTEX is considered a preferable option, due to its practicality [24].

The electron beam (EB) method is employed to create a high-energy electron beam through electron accelerators. High-energy electrons ionize and excite water vapor and oxygen molecules in the gas, providing active particles such as ·OH, ·O, and ·OH. Active particles then decompose BTEX via a series of oxidation reactions [25]. The EB method is especially suitable for low-concentration and large-volume BTEX control. Son et al., developed an electron beam/ultra-fine bubble (EB/UB) hybrid system to reduce the byproduct formation during EB radiation. The results show that the toluene (30 ppm) removal efficiency of the EB/UB hybrid system was 97.1%, 17% higher than that of EB only (10 kGy) [26]. Kim et al., compared the decomposition effect of o-xylene using EB-only and EB-catalyst coupling systems. When irradiating 1500 ppmC o-xylene at room temperature under a 10 kGy absorbed dose condition, the removal efficiency of an EB-catalyst for o-xylene reached 94.5%, which was twice as high as that with EB-only [27]. The photocatalytic method utilizes light-induced semiconductor production of ·OH to oxidize BTEX. Photocatalysis has high photoactivity and a rapid reaction speed [28,29,30]. Chen et al., synthesized a non-noble amorphous SnO_2_-modified ZnSn(OH)_6_(ZSH) photocatalyst by the in situ method. Compared to the original ZSH and noble-metal-modified ZSH, the toluene removal rate by ZnSn(OH)_6_(ZSH) was increased by 13.0 and 3.8 times, and the toluene mineralization rate was increased by 5.2 and 2.2 times [31]. Xue et al., compounded conductive Ti_3_C_2_ with TiO_2_ for photocatalytic degradation of gaseous o-xylene. The results showed that the photodegradation efficiency of the 1% Ti_3_C_2_-TiO_2_ composites increased from 64.7% to 77%, compared to pure TiO_2_, while the mineralization rate of gaseous o-xylene was about double that [32]. Zhang et al., studied the photocatalytic efficiency of different PT/TiO_2_ catalysts for m-xylene at different relative humidity levels and O_2_ contents. The results showed that the photocatalytic effect of m-xylene was the best at 21% O_2_ and 0% relative humidity, with a removal rate of 97.5% [33]. While the advanced oxidation processes described above are efficient at oxidizing BTEX from the air, these methods require the collection of indoor gases and the direct contact of polluted air with catalysts in order to remove BTEX. As a result, applying these technologies in complex interior structures with dispersed sources of pollution requires a complete air circulation system, which incurs enormous equipment costs [34]. In addition, regular maintenance of air-cleaning equipment and catalyst replacement will also increase operating costs. Therefore, these technologies are unsuitable for BTEX removal in newly built apartments, schools, and other scattered semi-enclosed spaces. The completed buildings were also unsuitable for a large-scale renovation project to remove BTEX. Moreover, to maintain good indoor air quality (IAQ), it is imperative to reduce BTEX pollution. Typically, the best way to control BTEX is to reduce or eliminate its sources [35]. However, the aforementioned control technologies cannot eliminate the source of contamination.

Chlorine dioxide (ClO_2_) has a strong oxidative capacity and can react with organic compounds to remove contaminants [36]. ClO_2_ has a wide range of application values, due to its fast response, long-lasting effect, broad spectrum, simple operation, and high flexibility in removing VOCs. Although long-term exposure to high concentrations of ClO_2_ can damage human cells, ClO_2_ will decompose under light and cannot exist stably in the air; therefore, ClO_2_ gas is less harmful to humans and the environment than chlorine gas. In addition, the molecular dynamic diameter of ClO_2_ is 0.36 nm, which is smaller than most VOCs (0.39–0.7 nm). It is speculated that gaseous ClO_2_ can penetrate the pores where pollutants are released, diffuse through the material, and react with unreleased contaminants. Therefore, ClO_2_ has a unique high permeability and can significantly improve IAQ by oxidizing the BTEX source and reducing the duration of BTEX release [37]. Moreover, gaseous ClO_2_ is convenient and can address the problem of BTEX contamination source dispersion in complex indoor structures. Its powerful oxidizing properties and high permeability allow ClO_2_ to be applied in antimicrobial decontamination in medical areas, food processing, and in odor mitigation [38]. Studies have shown that ClO_2_ has a pronounced ability to remove gaseous contaminants in semi-enclosed spaces. Wu et al., studied the degradation capacity of formaldehyde with different concentrations of ClO_2_. By comparing the removal capacity of 1 mg/m^3^ formaldehyde with 5 mg/m^3^, 25 mg/m^3^, 50 mg/m^3^, and 100 mg/m^3^, it was found that when the dosage of ClO_2_ was increased from 25 mg/m^3^ to 100 mg/m^3^, the maximum removal rate of formaldehyde increased from 60% to 90% in 60 min [39]. Ganiev et al., summarized the reaction rate constants of ClO_2_ with various organic compounds and found that ClO_2_ has the ability to oxidize many PAHs [40,41].

Given the cost-effectiveness and safety of BTEX removal, the key is to study the efficiency of ClO_2_ in oxidizing BTEX and prevent secondary contamination. However, the literature lacks the efficiency and mechanism for BTEX removal using ClO_2_. Furthermore, it is challenging to obtain evidence of the reaction mechanism from experiments, due to the lack of testing methods of the reaction intermediates [42]. Therefore, the primary purpose of this study is to examine the removal properties of liquid and gaseous benzene, toluene, o-xylene, and m-xylene with ClO_2_, to determine the optimized operating conditions when treating BTEX with ClO_2_ in semi-enclosed areas. Moreover, analysis-testing means and theoretical calculations were combined to analyze the byproducts of the reaction process, to discuss the mechanism of the reaction and to ensure the safety of the oxidative products.

## 2. Results and Discussion

### 2.1. Removal of Liquid BTEX by ClO_2_

The blank experiment was carried out under a sealed conical flask without adding ClO_2_. The results showed that the concentration of BTEX (10 mg/L) had not significantly changed in 10 h (the changing range was within 0.1%), and the influence on the experimental results was negligible. The blank experiment demonstrated the stability of BTEX.

Figure 1 shows the removal effect of ClO_2_ concentration on the liquid BTEX removal rate. Experimental results show that BTEX removal increases with increasing reaction time. In addition, as the concentration of ClO_2_ increases, the BTEX removal rate revels the same trend of first increasing, and then decreasing. Figure 1a shows the removal rate of 10 mg/L liquid benzene by 30 mg/L–50 mg/L ClO_2_. The optimal degradation rate of benzene at 5 h is 47.22% at 45 mg/L ClO_2_, which is 10.29% higher than that of 40 mg/L ClO_2_. Figure 1b shows the removal rate of 10 mg/L liquid toluene by 12 mg/L–36 mg/L ClO_2_. The oxidation efficiency of ClO_2_ on toluene is more pronounced than that of benzene. The highest removal rate for toluene was 42.77% at 30 mg/L ClO_2_ for 5 h. ClO_2_ concentration change has little effect on the toluene removal. When the concentration of ClO_2_ increased to 36 mg/L, the removal rate of toluene decreased by 4.44%. Figure 1c shows the removal rate of 10 mg/L liquid o-xylene by 0.1 mg/L-5 mg/L ClO_2_. The optimal ClO_2_ dosage required for o-xylene oxidation is only 0.5 mg/L, and the removal rate of o-xylene is 24.65%. Unlike toluene, ClO_2_ concentration change has a significant effect on o-xylene removal. With the increase of the ClO_2_ dosage from 0.5 mg/L to 5 mg/L, the removal rate of o-xylene gradually decreased. At a ClO_2_ concentration of 5 mg/L, the o-xylene removal rate dropped to 0.05%. This suggests that the optimal concentration of ClO_2_ for o-xylene is lower than that for benzene and toluene, due to the lower stability of o-xylene molecules. However, the low ClO_2_ dosage may lead to premature oxidant depletion, reducing the o-xylene removal rate. Figure 1d shows the removal rate of 10 mg/L liquid benzene by 3 mg/L–7 mg/L ClO_2_. Compared to the o-xylene molecule, the m-xylene molecule exhibits a more significant symmetry and is therefore more stable than the o-xylene molecule. This results in a higher ClO_2_ concentration required for the oxidation of m-xylene than o-xylene. The optimal ClO_2_ dosage required for removing m-xylene is 5 mg/L, under which the removal rate of m-xylene can reach 41.24% within 5 h. Additionally, the effect of ClO_2_ concentration on the removal rate of m-xylene was also significant. The removal rate of m-xylene decreased from 41.24% to 23.86% when ClO_2_ concentration increased by only 1 mg/L. These results indicate that ClO_2_ is effective in removing liquid BTEX as representative VOCs.

Figure 2 examined the removal rate of BTEX by optimal ClO_2_ dosage (45 mg/L, 30 mg/L, 0.5 mg/L, and 5 mg/L for benzene, toluene, o-xylene, and m-xylene, respectively) at 24 h. The experimental data show that the reaction rate increases slightly when using ClO_2_ to oxidize BTEX for more than 5 h. Compared with the result at 5 h, the removal rate of benzene, toluene, o-xylene, and m-xylene at 10 h increased by 9.8%, 26.89%, 3.7%, and 7.01%, respectively. However, when the reaction continued for 24 h, the effect of ClO_2_ removal on BTEX did not improve significantly, with removal rates increasing by 5.36%, 3.72%, 1.42%, and 2.65%, respectively. Thus, the optimal reaction time for oxidizing liquid BTEX with ClO_2_ is 10 h.

### 2.2. Removal of Gaseous BTEX by ClO_2_

The blank experiment was carried out under sealed and dark conditions, without adding ClO_2_. The results showed that the concentration of BTEX (4 mg/m^3^) had not significantly changed in 10 h (the changing range was within 0.1%), and the influence on the experimental results was negligible. The blank experiment demonstrated the stability of o-xylene and the reliability of the reactor.

Figure 3 shows the effect of ClO_2_ concentration on the gaseous BTEX removal rate. Experimental data reveal that the removal rate of gaseous benzene revels a trend of first increasing and then decreasing with increasing ClO_2_ concentration, which was the same as with liquid benzene. Figure 3a shows the removal rate of 4 mg/m^3^ gaseous benzene by 10–50 mg/m^3^ ClO_2_. The removal rate of benzene increases rapidly and then becomes stable after 2 h. When the ClO_2_ dosage was 25 mg/m^3^, the removal effect of benzene reached a maximum of 81.75% in 10 h. Figure 3b shows the removal rate of 4 mg/m^3^ gaseous toluene by 10–50 mg/m^3^ ClO_2_. The figure demonstrates that the initial rate of the reaction was fast during the first 2 h, then decreased significantly and remained virtually constant. The highest toluene removal rate of 72.04% was achieved by 20 mg/m^3^ ClO_2_ in 10 h. Figure 3c shows the removal rate of 4 mg/m^3^ gaseous o-xylene by 1–15 mg/m^3^ ClO_2_. The initial reaction rate is relatively fast for the first 2 h, after which it decreases markedly. The highest o-xylene removal rate (64.06%) was observed with 8 mg/m^3^ of ClO_2_ and 10 h. Figure 3d shows the removal rate of 4 mg/m^3^ gaseous m-xylene by 40–120 mg/m^3^ ClO_2_. For the first 2 h, m-xylene reacts with ClO_2_ at a relatively rapid rate. However, the removal rate of m-xylene within 2 h is about 15% lower than that of benzene, toluene, and o-xylene. After 2 h, the removal rate tends to rise slowly. The removal rate of m-xylene reached its highest level of 57.49% at a concentration of 80 mg/m^3^ and 10 h of ClO_2_.

In order to study the effect of ClO_2_ removal on mixed BTEX gas, 10–90 mg/m^3^ ClO_2_ was used to oxidize 4 mg/m^3^ each of benzene, toluene, o-xylene, and m-xylene as a gaseous mixture, as shown in Figure 4. From the data, it is evident that the BTEX removal increases with the reaction time. Similarly to the BTEX removal experiment studied above, the mixed gaseous BTEX reacts rapidly with ClO_2_ within 2 h, followed by a significant slowing of the reaction rate after 2 h. This phenomenon shows that the optimal reaction time for oxidizing gaseous BTEX with ClO_2_ is 2 h. In contrast to the results obtained using ClO_2_ to oxidize a single gaseous BTEX, the optimal ClO_2_ dosage and the optimal removal rate of mixed gaseous BTEX at 10 h change significantly, due to multiple BTEX interactions. Figure 5 shows the removal rate of a 4 mg/m^3^ gaseous mixture of benzene, toluene, o-xylene, and m-xylene by 10–90 mg/m^3^ ClO_2_ in 10 h. It can be observed that the removal rates of benzene and toluene first increase and then decrease, with increasing ClO_2_ concentration. Additionally, 10 mg/m^3^ ClO_2_ has a higher removal effect on o-xylene and m-xylene than 30 mg/m^3^. After the ClO_2_ dosage exceeds 30 mg/m^3^, the removal rates of o-xylene and m-xylene first increase and then decrease with increasing ClO_2_ concentration. When ClO_2_ concentration was 50 mg/m^3^, the removal rates of benzene, toluene, and p-xylene reached the highest, which were 53.09%, 61.70%, and 61.27%, respectively. Under this condition, the removal rate of o-xylene was 58.54%, which was 1.08% lower than that of the optimal ClO_2_ dosage (10 mg/m^3^). Combined with the outcomes obtained from o-xylene oxidation by ClO_2_, this phenomenon is mainly due to the low optimal ClO_2_ concentration required for o-xylene oxidation. The results showed that the optimal ClO_2_ concentration of 4 mg/m^3^ mixed BTEX was 50 mg/m^3^.

### 2.3. BTEX Removal Mechanism

The above results confirmed that ClO_2_ has a strong ability to remove both liquid and gaseous BTEX. However, Ganiev et al., reported that ClO_2_ does not react directly with the simplest alkyl-substituted benzenes such as benzene, toluene, and xylene, which was inconsistent with the above data [40]. However, studies have revealed that not only ClO_2_, which has an oxidizing effect on contaminants, but also ·OH radicals, generated during the reaction, contribute significantly to the oxidation process together. Sun et al., performed the NO removal from fuel gas with ClO_2_ and found that ClO_2_ has an excellent oxidizing effect on NO. The research also demonstrated, through the electron spin resonance (ESR) and inhibitor (sodium formate and isopropanol), the presence of ·OH in the reaction process, and it has been concluded that ·OH had a significant effect on the removal of NO [43]. Marcon et al., investigated the mechanism of the formation of radicals during ClO_2_ decomposition. The results of electron paramagnetic resonance (EPR) spectroscopy revealed the presence of ·OH with ClO_2_. Only a small amount of ·OH was detected with ClO^−^ at the same condition, indicating that liquid ClO^−^ reduced the production of ·OH. [44] According to research, ·OH is produced by the reaction of ClO_2_ with H_2_O (Equations (1)–(7)). [44,45,46,47] Moreover, it is clear that ·OH is efficient in degrading benzene, toluene, and xylene [48,49,50,51,52]. Therefore, it could be speculated that the ·OH produced by ClO_2_ with H_2_O plays a vital role in removing BTEX.
(1)$${\mathrm{ClO}}_{2}\cdot +{\;H}_{2}O↔{\mathrm{HClO}}_{2}+\cdot \mathrm{OH}$$
(2)$${\mathrm{ClO}}_{2}\cdot +\cdot \mathrm{OH}↔\mathrm{HOOClO}$$
(3)$${\mathrm{ClO}}_{2}^{-}+\cdot \mathrm{OH}↔{\mathrm{ClO}}_{2}+{\mathrm{OH}}^{-}$$
(4)$$\mathrm{ClO}\cdot +\cdot \mathrm{OH}↔{\mathrm{HClO}}_{2}$$
(5)$${\mathrm{ClO}}_{2}\cdot +{\;\mathrm{ClO}}_{2}\cdot ↔{\mathrm{Cl}}_{2}O_{4}$$
(6)$$\mathrm{HOOClO}↔H^{+}+{\mathrm{OOClO}}^{-}$$
(7)$${\mathrm{OOClO}}^{-}↔\mathrm{ClO}\cdot +\cdot O_{2}^{-}$$

ClO_2_ is soluble in water and exists in a stable dynamic equilibrium state. From a macroscopic point of view, ·OH, the intermediate product of the above disproportionation reaction, undergoes continuous production and disappearance. When adding BTEX to liquid and gaseous ClO_2_, ·OH (2.8 eV), which has a higher redox potential than ClO_2_ (1.5 eV), will rapidly oxidize BTEX. The reaction not only removes BTEX but also considerably promotes the forward progress of Equation (1) [43]. Thus, a further increase in ClO_2_ would reduce the speed of Equation (1) and promote the forward progress of Equations (2)–(4), resulting in a decrease in the ·OH concentration and a rapid reduction in the BTEX removal rate. The decrease in ·OH is one of the reasons why the BTEX removal rate decreases at high ClO_2_ concentrations, which is in agreement with our experimental results in Figure 1 and Figure 3. Additionally, the presence of ·OH in the oxidation system may promote the production of various free radicals, which will interact and further enhance the oxidizing ability of ClO_2_ [43].

In order to analyze the oxidation byproducts of gaseous BTEX by ClO_2_, the gas chromatography-mass spectrometry (GC-MS) experiments were performed on gas samples from the reactor at 0 h, 0.5 h, 1 h, 4 h, and 10 h (Figure 6, Table 1). Figure 6a indicates a trace of organic compounds, including dibutyl phthalate, n-hexadecane, n-heptadecane, n-heptadecyl acetate, and methyl palmitate were formed from 0.5–10 h during the oxidation of benzene. Figure 6b shows that n-hexadecane, n-heptadecane, n-heptadecyl acetate, and methyl palmitate were the main products from 0.5–10 h, during the oxidation of toluene. Figure 6c,d indicate that the concentrations of o-xylene and m-xylene decreased significantly with the increasing reaction time, while no representative intermediates were produced. It can be inferred from this phenomenon that most o-xylene and m-xylene are mineralized into CO_2_ and H_2_O, and that some of the resulting intermediates have concentrations below the detection limit, in agreement with the results of Zou et al. [24].

Different BTEX degradation experiments reported different intermediates. d’Hennezel et al., reported that the main byproducts of benzene, under the action of ·OH, were phenol, phenylcatechol, and 1,4-benzoquinone. In contrast, Jacoby et al., reported that the photocatalytic oxidation of benzene produced phenol, malic acid, hydroquinone, benzoic acid, and benzoquinone [53,54,55]. He et al., speculated that ·OH replaced an ·H molecule of benzene to produce phenol. After the ring-opening process, 1,3-butanediol, glycolic acid, and other small molecule alkanes were produced and finally mineralized into CO_2_ and H_2_O [56]. Zhong et al., detected the presence of ethyl acetate, 2,5-cyclohexadiene1,4,diketone,2,6-bis(1,1-dimethyl), butylated hydroxytoluene, and dibutyl phthalate from the GC-MS results of the photocatalytic degradation of gaseous benzene. In addition, it reveals that the long-chain alkanes produced by the ring-opening products further react with benzene and ·OH to produce dibutyl phthalate [57]. Based on the GC-MS results above, the oxidation mechanism of gaseous BTEX by ClO_2_ was speculated and depicted in Figure 1. The oxidation of benzene, via the attack of the six-membered ring with ·OH, produces phenol [48]. The phenol can be oxidized by ·OH to form intermediates such as phenol and benzoquinone. These byproducts further degrade into ring-opened species, such as acetaldehyde, 2-butanone, and other fragmented alkyl groups [56]. Most of these products will mineralize to form CO_2_ and H_2_O, while some will produce species with higher carbon numbers, such as n-hexadecane, n-heptadecyl acetate, and methyl palmitate [57]. The oxidation of toluene is possible, due to the attack of the methyl with ·OH, which produces benzaldehyde and benzoic acid. The phenols then undergo a ring-opening reaction to produce CO_2_, H_2_O, and species with higher carbon numbers [58]. Another possible pathway for toluene is through the oxidation of ·OH, which leads to the break-off of the methyl group on the toluene to produce benzene, which is then further oxidized and decomposed [59]. There are two possible pathways for the degradation of o-xylene and m-xylene: (1) through the attack of the methyl with ·OH to form phenols such as o-quinone and m-quinone [60,61,62,63], and (2) through the break-off of the methyl group to produce toluene [64]. After the ring-opening process, most of the product will eventually mineralize into H_2_O and CO_2_ [60,65,66]. ·OH attacks the six-membered ring of BTEX, which is a key factor in the decomposition and removal of BTEX in semi-enclosed spaces. Therefore, the translation states of the degradation pathway of benzene (Figure 7), the most stable compound of BTEX, with ·OH, was calculated to verify the reliability of the reaction pathway. The transition state and the relative thermodynamic properties of each reaction are computed as follows:(8)$$\Delta G_{r}=G_{\left(X\right)}-G_{ref}$$
in which *G_(X)_* and *G_ref_* are the Gibbs free energy of structure *X*, and the reference, respectively (Figure 8 and Table 2). The first step of each reaction is considered as the reference. The results show that, at 298.15 K and 1 atm, the initial relative Gibbs free energy of the ·OH-initiated reaction (AA* and BB*) is 96.24 KJ/mol and 110.32 KJ/mol, respectively, indicating that gaseous benzene will react with ·OH to produce catechol. Catechol is known to yield quinones in a water solution [57,67]. Morales-Roque et al., found that the degradation of catechol depends on the presence of ·O_2_^−^ radicals that also exist in the disproportionation reaction of ClO_2_ (Equation (7)) [68]. This feature of the reaction mechanism cannot be explained thoroughly, in view of the effect of ·OH, and requires a more precise investigation of the energy values of the reaction intermediates. This is the reason for the high initial relative Gibbs free energy of the reaction of catechol with ·OH (CD and DE, 214.75 KJ/mol and 209.07 KJ/mol, respectively). The research confirms that ·OH is able to react with BTEX in semi-enclosed areas. Furthermore, Ben Amor et al., have reported that o-quinone can undergo a ring-opening reaction under the oxidation of ClO_2_ [69].

## 3. Materials and Methods

### 3.1. Experimental Materials and Apparatus

Analytical grade (≥99%) liquid primary standards of benzene, o-xylene, m-xylene, and Carbon disulfide were purchased from Shanghai Macklin Biochemical Co., Ltd. (Shanghai, China). Toluene (≥99%) and Sodium chlorite (≥80%) was purchased from China National Medicines Corporation Ltd. (Beijing, China). Sulfuric acid (≥98%) was purchased from China Sinopharm International Co., Ltd. (Shanghai, China).

The ClO_2_ stock solution in this experiment was prepared daily, according to the standard method [70]. A total of 20 mL of sulfuric acid (10%) was slowly added into 0.013 g/mL of sodium chlorite solution, and the resulting gaseous ClO_2_ was discharged into a borosilicate glass collection bottle. The content of ClO_2_ was determined by an ultraviolet (UV) spectrophotometer (UV-8000S, Metash Co., Ltd, Shanghai, China) and the five-step iodometric method. The characteristic UV absorption peak of ClO_2_ is at 430 nm, and the ClO_2_ was quantitative when using the external standard method. The prepared ClO_2_ solution was placed in a sealed amber glass bottle and stored away from light.

Experiments were performed using a sealed reactor with a volume of 200 L (Figure 9). The reactants were added from the injection port (1). The reactor was equipped with two fans (2, 3) to ensure the volatilization of BTEX. Samples were withdrawn from the sampling point (3) at the time for analysis.

### 3.2. BTEX Removal Experiments

The liquid BTEX removal experiment was performed in a 250 mL sealed conical flask. 10 mg/L of BTEX solution and different concentrations of liquid ClO_2_ were added to the flask at room temperature. A magnetic stirrer ensured complete mixing. Samples were taken every 1 h and then analyzed immediately using high-performance liquid chromatography (HPLC). The BTEX concentration was calculated using the external standard method. Each process was repeated three times to ensure the reliability and repeatability of the results.

The decomposition ratio (*η*) of liquid BTEX was calculated as
(9)$$\eta =\left(1-\frac{C_{t}}{C_{0}}\right)\times 100\%$$
where *C*_0_ and *C_t_* depict the concentration of liquid BTEX at 0 h and *t* h, respectively.

The gaseous BTEX removal experiments were performed in the reactor mentioned above. The dosage of BTEX refers to the concentration of BTEX detected in the air of unventilated renovated houses. A total of 4 mg/m^3^ of benzene, toluene, o-xylene, and m-xylene was added to the reactor at room temperature, respectively. The fans were turned on to completely mix the gaseous BTEX. After an hour, a sample was taken as blank control to reduce experimental errors. The effects of ClO_2_ concentration and reaction time were investigated experimentally. The sampling and detection of benzene series are based on the standard HJ 584-2010: Ambient air—Determination of benzene and its analogies by activated charcoal adsorption carbon disulfide desorption and gas chromatography [71]. The products were collected by a 150 mg activated carbon tube (HPLC/ACS, Macklin Biochemical Co., Ltd, Shanghai, China) at a flow rate of 0.6 L/min, desorbed with 1 mL carbon disulfide solution for 1 h, and then analyzed using gas chromatography (GC) with a flame ionization detector (FID). BTEX was identified through its retention time and was quantitative, using the external standard method [72]. Each process was repeated three times to ensure the reliability and repeatability of the results.

The decomposition ratio (*η*) of gaseous BTEX was calculated as
(10)$$\eta =\left(1-\frac{C_{t}}{C_{0}}\right)\times 100\%$$
where *C*_0_ and *C_t_* depict the concentration of gaseous BTEX at 0 h and *t* h, respectively.

### 3.3. Analytical Methods

All liquid BTEX samples were analyzed using a Thermo Scientific UltiMate 3000 HPLC with a UV detector. Separations were carried out using a thermostated (31.6 °C) 150 mm × 4.6 mm, 5 μm Thermo Hypurity C_18_ column, with methanol and water (70:30) used as mobile phase, and the flow rate was 1.0 mL/min. The UV detection wavelength was set at 254 nm. All aqueous solvents were filtered through 0.45 μm membrane filters.

All gaseous BTEX samples were analyzed using a Techcomp GC-7900 with a PEG-20M column (30 m × 0.32 nm × 1.00 μm) and an FID detector. The nitrogen carrier gas and makeup gas flow rates were 2.6 mL/min and 30 mL/min, respectively. The injector temperature was 150 °C, and the detector temperature was 250 °C. The column temperature rose in line with the program with an initial temperature of 65 °C, maintained for 10 min, and rose at a rate of 5 °C/min to 90 °C and was held for 2 min.

All reaction products were analyzed using a Bruker EVOQ GC-TQ with HP-5MS column (30 m × 0.25 nm × 0.25 μm). The Helium carrier gas flow rate was 1 mL/min. The ion source temperature was 230 °C. The column temperature rose in line with the program, with an initial temperature of 55 °C maintained for 3.2 min, then rose at a rate of 15 °C/min to 110 °C, was held for 1 min, and then rose to 180 °C at 30 °C/min and was held for 4 min.

### 3.4. Calculation Methods

The thermodynamic properties of benzene and its byproducts have been computed using the Gaussian 09W program package [73]. The ab initio molecular orbital calculations (MP2) method, in combination with the 6-31G (2d, p) basis set, has been used to obtain the geometric optimization of all reactants, products, and transition states [74]. The vibrational frequencies were performed in the same level of theory for the optimized reactants, products, and transition states to characterize transition states (N_imag_ = 1). The IRCs were calculated at a level of MP2/6-31G (2d,p) to ensure that the identified transition states smoothly connected the reactants and products [75]. To obtain more accurate energies, single point calculations were performed with the MP2 method with the larger def2-tzvpp basis set. Shermo software was used to calculate the thermodynamic variables at 1 atm and 298.15 K [76].

## 4. Conclusions

This paper verified that ClO_2_ is a convenient and effective method to remove liquid and gaseous benzene, toluene, o-xylene, and m-xylene. For 4 mg/m^3^ mixed BTEX gas, 50 mg/m^3^ ClO_2_ ensures the efficient removal of BTEX. With this ClO_2_ dosage, the removal rates of benzene, toluene, o-xylene, and m-xylene reach 53.09%, 61.70%, 58.54%, and 61.27%, respectively. Moreover, GC-MS analysis detected small concentrations of n-hexadecane, n-heptadecane, n-heptadecyl acetate, and methyl palmitate in the oxidation products of benzene and toluene. No representative intermediates were detected in the GC-MS results of the oxidation products of o-xylene and m-xylene. It is suggested that the ·OH, produced by ClO_2_ and H_2_O, played a significant role in the BTEX removal. Based on the results, the degradation mechanisms were speculated. This study illustrated the potential of ClO_2_ for indoor air pollution removal, and provided an effective method to exenterate BTEX in semi-closed spaces and to improve the IAQ. Since high concentrations of ClO_2_ can cause harm to humans, we recommend using ClO_2_ at night or when the buildings are empty.

## Data Availability

Not applicable.

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
