# Peer review of "Performance and Mechanism of Chlorine Dioxide on BTEX Removal in Liquid and Indoor Air"

_molecules, 2023, doi:10.3390/molecules28114342_

Round 1

Reviewer 1 Report

I think the study is interesting but has a serious default, that is the use of a compound more dangerous to decrease the levels of compounds less dangerous. I think the study is valid, but this methodology can not be used in spaces with people. Perhaps in places like archives, for example. But it is important that the authors emphasize this aspect of the possible health effects.
Next, I leave some suggestions, which I hope will help to improve the article. In all the text it is used the abbreviation BETX (benzene, ethylbenzene, toluene, and xylenes), but normally it is used BTEX. I think it's best to change that throughout the text. It is written: “The adsorption method is to separate and remove BETX pollution by adsorbing the contaminant on the surface of the adsorbent.” – I think this sentence doesn't sound right and the word “separate” doesn’t apply. Please rephrase the sentence, for example: “The adsorption method consists in removing BETX pollution by adsorbing the contaminants on the surface of the adsorbent.” It is written: “At room temperature, 50 mg hexadecyltrimethoxysilane (HDTMS)-WSN can thoroughly remove 200 μg/m benzene, toluene, and xylene mixed air samples [13].” – You want to say 200 ug/m3? Please correct it.

It is written: “The electron beam (EB) method is to accelerate the electrons produced by electron generation devices through electron accelerators.” – I think this explanation is a little “circular”. Please rewrite it.
It is written: “ClO2 has a wide range of application values due to its fast response, long-lasting effect, broad-spectrum, safe without side effects” – I can not agree with this sentence. Please see the link https://limitvalue.ifa.dguv.de – you can see that ClO2 is not an innocuous compound. The limit value for occupational exposition is around 0.3 mg/m3! If you read the article “Virucidal, bactericidal, and sporicidal multilevel antimicrobial HEPA-ClO2 filter for air disinfection in a palliative care facility” (https://doi.org/10.1016/j.cej.2021.134115) you can see that ClO2 can damage the human cells. This compound is also bad for the environment. So, you can not ignore this. Please correct the sentence, and add some precautional advices on use this product. It can not be used indoors, with people. Perhaps only at night, when the buildings are empty. It is written: “The optimal degradation rate of benzene at 5 h is 47.22% at 45 mg/L ClO2, which is 10.29% higher than that of 45 mg/L ClO2.” – There is a mistake in this sentence. Please correct it. It is written: “Similarly to the BETX removal experiment studied above, the mixed gaseous BETX reacts rapidly with ClO2 within 2 h, followed by a significant slowing of the reaction rate after 2 h.” – This means that the optimal reaction time for oxidizing gaseous BETX with ClO2 is 2 h? Please add this information in the text. Another relevant question is what remains of ClO2. What is the final concentration of ClO2 in the air at the end of the optimal reaction time? It is all consumed? It is written: “The ClO2 stock solution in this experiment was prepared daily according to the standard method [70]. Dilute H2SO4 was added to a sodium chlorite solution,…” – add details about reagents like H2SO4 and sodium chlorite. It is written: “The sampling and detection of benzene series are based on the Ambient air — Determination of benzene and its analogies by activated charcoal adsorption carbon disulfide desorption and gas chromatography [71].” – this is a standard, so I think the authors should add this information in the sentence, as for example: “The sampling and detection of benzene series are based on the standard HJ 584-2010: Ambient air — Determination of benzene and its analogies by activated charcoal adsorption carbon disulfide desorption and gas chromatography [71].” In the Conclusions it is written: “This study illustrated the potential of ClO2 for indoor air pollution removal and provided a safe and effective method to exenterate BETX in semi-closed spaces and to improve the IAQ.” – please rewrite based on what I have said before about the danger of using ClO2 indoors.  

Reviewer 2 Report

This study illustrated the potential of chlorine dioxide for indoor air pollution removal and provided a safe and effective method to exenterate BETX in semi-closed spaces. It is innovative and should be accepted after minor revision.

1. The introduction needs to be simplified to highlight the innovation of the article.

2. For the liquid BETX removal, what is the solvent of BETX? Does it affect the action of chlorine dioxide?

3. The figure symbols are too small to see clearly. The quality of the picture needs to be improved.

4. The experimental data in Fig. 2a didn’t show that the reaction rate decreases with the increasing reaction time, but increases slightly.

5. EPR and quenching experiments should be supplemented to further demonstrate the generation of ·OH and ·OH plays a vital role in removing BETX.

6. The hydroxyl radicals (·OH) in some places of the article are written incorrectly, please check and modify them.

7. It is necessary to further determine the final destination of chlorine dioxide, and whether harmful chlorite will be generated, causing secondary pollution.

English should be improved.

Round 2

Reviewer 1 Report

I think the authors have responded adequately to requests, so I think the article can be accepted.

Reviewer 2 Report

The quality became better after the author made more effort. The revised manuscript is suitable for publishing in molecules.